# Study on Material Design and Corrosion Resistance Based on Multi-Principal Component Alloying Theory

**DOI:** 10.3390/ma16051939

**Published:** 2023-02-26

**Authors:** Beiyi Ma, Hongyang Zhao, Dongying Ju, Zhibo Yang, Ming Chen, Qian Liu

**Affiliations:** 1School of Mechanical Engineering and Automation, University of Science and Technology Liaoning, Anshan 114051, China; 2School of Materials and Metallurgy, University of Science and Technology Liaoning, Anshan 114051, China; 3Saitama Institute of Technology, Fukaya 369-0203, Japan

**Keywords:** corrosion resistance, bio-materials, Mg_30_Zn_30_Sn_30_Sr_5_Bi_5_ alloy, electrochemical corrosion test, multi-principal alloying

## Abstract

This study mainly attempts to develop Mg-based alloy materials with excellent corrosion resistance by means of multi-principal alloying. The alloy elements are determined based on the multi-principal alloy elements and the performance requirements of the components of biomaterials. Mg_30_Zn_30_Sn_30_Sr_5_Bi_5_ alloy was successfully prepared by vacuum magnetic levitation melting. Through the electrochemical corrosion test with m-SBF solution (pH7.4) as the electrolyte, the corrosion rate of alloy Mg_30_Zn_30_Sn_30_Sr_5_Bi_5_ alloy decreased to 20% of pure Mg. It could also be seen from the polarization curve that when the self-corrosion current density is low, the alloy shows superior corrosion resistance. Nevertheless, with the increase in self-corrosion current density, although the anodic corrosion performance of the alloy is obviously better than that of pure Mg, the cathode shows the opposite situation. The Nyquist diagram shows that the self-corrosion potential of the alloy is much higher than that of pure Mg. In general, under the condition of low self-corrosion current density, the alloy materials display excellent corrosion resistance. It is proved that the multi-principal alloying method is of positive significance for improving the corrosion resistance of Mg alloys.

## 1. Introduction

Biomaterials are a new type of material used to repair or replace damaged tissues and organs in the process of biological therapy [1]. Their basic properties include bio-compatibility, corrosion resistance, reversibility of degradation, appropriate elastic modulus, and fatigue strength [2]. With the continuous progress and development of society, medical means have further developed from simple treatment of diseases to better treatment experiences for patients at the same time. At present, many implant materials have been clinically applied, such as cobalt-based alloy, titanium-based alloy, stainless steel, etc. However, there is a common problem with these materials, that is, “they need to be removed by a second operation and produce metal ions that are harmful to human health during treatment” [3]. By contrast, because of its unique degradability and safety, Mg alloy bio-materials effectively avoid the above problems, and therefore have become an important research topic in the field of new medical materials. As a biodegradable biomaterial, the performance advantages of magnesium and its alloys mainly include the following [4]: (1) In terms of comprehensive mechanical properties, the elastic modulus (41~45 GPa) of Mg and Mg alloys is closer to the elastic modulus of biological bones than other metal materials, which can effectively avoid the problem of osteogenesis imperfection caused by “stress shielding”, and reduce the discomfort of patients caused by the implantation of “hard” metals in the body; (2) as important trace elements of the human body, the daily intake of which can reach 240~420 mg for adults, Mg and Mg alloys are safe and harmless to human body even if corrosion reaction occurs after implantation, so they show better bio-compatibility; and (3) Mg and Mg alloys, as biological implant materials, show good biodegradability after implantation, which effectively avoids the problem that other metals must be “removed by secondary surgery” [5]. They also have anti-inflammatory, anti-tumor, and antibacterial properties, which can reduce the rejection and pathological changes in patients due to the implantation of foreign objects. At the same time, its excellent osteoinductive properties can stimulate the growth of bone cells and ensure the effective recovery of bone function. Based on the above characteristics, Mg and Mg alloys are considered degradable implant materials with great research value and development prospects [6].

As degradable biomaterials, Mg and Mg alloys have received widespread attention in the research field, but the rapid degradation rate has become the biggest obstacle to their transformation from “laboratory materials” to “practical materials”. Therefore, how to improve the corrosion resistance and control the degradation rate of Mg and Mg alloys has become a key point in the research of biological Mg materials. The methods of adjusting the corrosion resistance of Mg-based alloys have been explored in the research of Mg-based biomaterials, such as alloying, surface treatment, amorphization, and composite materials. Fintováa [7] et al. immersed AZ61 Mg alloy in molten salt of Na[BF_4_] to prepare fluoride conversion film to improve its corrosion resistance. Jeong [8] et al. compared the casting with different Ca content and the degradation of extruded Mg-Ca alloys in Hank’s simulated body fluid solution, and proved that the corrosion resistance of the Mg-Ca alloys was greatly enhanced in the Mg-Ca alloys with Ca ≥ 1% after extrusion. Mohamed [9] et al. compared the degradation rate of Mg-0.8Ca and pure Mg in Hank’s simulated body fluid solution. The results showed that the Mg-0.8Ca alloy had an approximately 3-fold faster degradation rate than the pure Mg. Koc [10] et al. compared the electrochemical degradation rate of as-cast Mg xZn alloy in a simulated body fluid solution, and proved that the degradation rate of the alloy decreased with the increase in Zn content. Zhao [11] et al. compared the degradation rate of extruded Mg xSr alloys with different Zn content in Hank’s simulated body fluid, and proved that the degradation rate of Mg xSr alloys increased with the increase in Sr content. Tian et al. [12] compared the degradation behavior of Mg-2Zn-0.2Mn-1Ca alloys in simulated body fluid under different heat treatment conditions. The SEM results showed that with the increase in heat treatment time, the degradation rate first decreased and then increased, precipitation amount decreased, and grain size increased. Janbozorgi [13] et al. compared the degradation performance of as-cast and solid solution Mg-2Zn-1Gd-1Ca alloys in simulated body fluid solution. The results showed that the dissolution of a large number of precipitates inhibited galvanic corrosion to a certain extent, which ultimately led to a significant decrease in the alloy’s degradation rate after solution treatment. Zhong [14] et al. compared the degradation characteristics of Mg-8Sn-2Zn-0.2Mn alloys with different pre-treatment duration in 3.5% NaCl solution, which strongly proved that the degradation properties of alloys can be adjusted by different aging treatments. Liu [15] et al. compared the degradation characteristics of as-cast Mg-Gd-Zn-Zr and heat-treated one. They found that the degradation of the alloy was improved because the reduction in the potential difference can weaken galvanic corrosion. Torroni [16] et al. analyzed the cast WE43 alloy and the heat-treated ones implanted into sheep skulls. The results showed that compared with the cast WE43 alloy, the heat-treated alloy had better stability and a lower degradation rate.

In this study, a new method was adopted to develop a biodegradable Mg alloy biomaterial with better corrosion resistance by referring to the design of high entropy alloys and increasing the proportion of alloy principal components. This study attempts to develop a degradable Mg alloy biomaterial with better corrosion resistance by the multi-principal alloying method. With the continuous development of material science, the literature on binary and ternary alloys has been well established. However, the research on multi-principal alloys provides a new possibility and is expected to break this limitation. High entropy alloy is a typical representative of multi-principal alloys. According to the traditional alloy design concept, the larger the number of main elements is, the easier it is to form complex phases, such as intermetallic compounds, which will lead to a serious decline in alloy performance. However, in the design of the multi-main element materials, it is believed that although it has multiple main elements, it can form a single solid solution structure, thus obtaining four core effects, that is, high entropy effect, lattice distortion effect, delayed diffusion effect and cocktail effect [17,18]. Multi-principal element materials with these four core effects are generally considered high-quality alloy materials with many superior properties, such as high strength, high hardness, good wear resistance, corrosion resistance, and fatigue resistance [19].

The design and preparation of Mg alloys involved in this study have been discussed and described in the paper Design, Simulation, and Performance Research of New Biometric Mg_30_Zn_30_Sn_30_Sr_5_Bi_5_, and has been approved the patent “*Mg-based Biomaterials based on the theory of superna alloy*, *their preparation methods and applications*”, patent number: *ZL 2021 1 1500334.1*. This study mainly focuses on electrochemical corrosion tests on the new material Mg_30_Zn_30_Sn_30_Sr_5_Bi_5_ and pure Mg. It is preliminarily proved that the corrosion resistance of the material was significantly improved due to the adjustment of the proportion of principal components.

## 2. Materials and Methods

### 2.1. Design Philosophy

As a new type of Mg alloy biomaterial, the material Mg_30_Zn_30_Sn_30_Sr_5_Bi_5_ alloy for corrosion performance test in this study is a multi-component alloy material. The element proportion standard is based on the multi-principal element alloying theory. The element selection is on the basis of implantability of organisms. The standards and bases are shown in Table 1.

Combined with the contents of Table 1 and Figure 1 with the element Mg as the research subject, the results obtained are shown in Table 2.

### 2.2. Alloy Fabrication

As the selected materials Mg, Zn, Sn, Sr, and Bi are all active metal elements, they are easy to be oxidized in the smelting process, making the loss rate high and the preparation difficult. Therefore, the more advanced vacuum electromagnetic levitation melting method was employed to prepare the material in this study. Vacuum electromagnetic levitation melting makes the molten alloy suspend in the crucible through electromagnetic levitation technology, so as to ensure that the material would not contact the crucible during metal smelting. This can avoid not only unnecessary chemical reactions between the molten metal and the elements contained in the crucible, but also the pollution of the crucible materials to the molten pool. Thus it can produce high-purity materials with more uniform composition. Vacuum electromagnetic levitation smelting is very suitable for active metal smelting. Raw materials used for smelting metal are shown in Table 3. Vacuum electromagnetic levitation melting operation was completed by Beijing Yijin New Material Technology Co., Ltd. (Beijing, China, https://yijinxc.cn.china.cn, accessed on 28 June 2022). The manufacturer of vacuum induction electromagnetic levitation smelting furnace is Jinzhou Taihe District Weili Metallurgical Equipment Factory (http://www.jzwlyj.cn/index.php/cn/Index/index.html, accessed on 28 June 2022), Equipment model: *ZG2-XF*, Capacity: 2 kg, Mg_30_Zn_30_Sn_30_Sr_5_Bi_5_ alloy weight: 1 kg.

Melting steps:

Step 1. Clean the raw materials to remove impurities and oxide layers on the surface. Then, store the cleaned materials in vacuum until they are mixed in proportion and put into the furnace.

Step 2. Vacuum the furnace. When the vacuum degree reaches 10^−3^ MPa, clean the gas in the furnace with argon twice. After that, fill argon into the whole furnace to ensure that all smelting and casting processes are conducted under argon protection.

Step 3. Adopt the step mode for the increase in smelting current. The current values are 100 Kw–150 kW–200 kW in turn, with a time interval of 5 min. When all raw materials are melted, keep the smelting current at 150 kW for 15 min. The total smelting time is about 30 min.

Step 4. After melting, take the alloy out of the furnace when it is cooled to 100 °C.

The smelting process was repeated 4 times, and the alloy solution was turned up and down each time. The above steps were repeated before the smelting of new materials was completed.

### 2.3. Corrosion Tests

Square samples with dimensions of 15 mm × 15 mm × 3 mm were cut from Mg_30_Zn_30_Sn_30_Sr_5_Bi_5_ alloy and pure Mg to measure the corrosion rate. The test instrument was GAMRY INTERFACE 1000E (Gamry Instruments, Warminster, PA, USA). The samples were ground and polished with sandpaper and 1.5-micron diamond polishing paste, and were washed with ethanol and dried. Three-electrode system was adopted, the reference electrode was saturated calomel solution, and the auxiliary electrode was platinum electrode. The working electrode was the tested sample with an effective area of 1 cm^2^. The corrosive medium used was m-SBF (pH 7.4) solution. The solution was not stirred during the immersion as well as electrochemical corrosion test [38]. m-SBF (pH 7.4) was a modified simulated body fluid. Its difference from traditional simulated body fluid is shown in Table 4. It can be seen from the table that compared with the traditional SBF solution, the ions contained in the m-SBF (pH 7.4) solution are closer to the concentration of various ions in the plasma, and can almost achieve 100% restoration, which can make the in vitro simulation results more consistent with those of the in vivo simulation. Test process: First of all, materials were exposed to the solution for 0.5 h in the open circuit potential (OCP) to stabilize the surface. Then, electrochemical impedance spectroscopy (EIS) was performed from 10^−2^ Hz to 10^5^ Hz to draw the Nyquist and Bode plots. Finally, potentiodynamic study was performed in the range of −1 V to 1 V with respect to OCP with scan rate of 1 mV/s and step height of 5 mV.

### 2.4. Phase Identification and Microstructure

The phases presented in Mg_30_Zn_30_Sn_30_Sr_5_Bi_5_ alloy were evaluated by XRD spectrum analysis, and were obtained by X-ray diffractometer (Rigaku Smartabl 9, Rigaku Corporation, Tokyo, Japan) with Cu Kα radiation at 40 kV and 20 mA. The scan speed of 10°/min and a step size of 0.02° over the 2θ range of 5° to 90° were employed during the scanning. SEM equipped with EDS (ZEISS GeminiSEM 300, ZEISS, Jena, Germany) was used to observe the surface morphologies of Mg_30_Zn_30_Sn_30_Sr_5_Bi_5_ and to detect the presence of elements in all the samples [39].

### 2.5. Theoretical Calculation of Alloy Properties

Based on the fact that there are many uncontrollable factors in the research of new materials, the calculation of related properties was carried out using the CALPHAD Technology, which is well developed at present; therefore, the feasibility of the research was confirmed. The research mainly focused on thermodynamic properties and solidification phases. The theoretical formulas include the thermodynamic principles, the Scheil–Gulliver Model, and the calculation of material properties. This part has been discussed in published papers [40].

## 3. Discussion

### 3.1. Corrosion and Degradation Mechanism of Mg and Mg Alloys

The normal temperature of the human body is 37 °C, and the pH value is basically maintained at 7.4, which is generally weak alkaline. The weak alkaline environment is very harmful to the metal hydride protective layer, and can accelerate the corrosion of metal materials [41]. The internal environment of organisms is very complex and contains sodium ions, chloride ions, calcium ions, bicarbonate anions, and other inorganic salts [42]. These elements have various effects on the corrosion degradation process of Mg alloys. Therefore, the degradation process of Mg alloy in simulated body fluid is a very complex process. For example, there are a large number of Cl^−^ ions in the body fluid, and Cl^−^ ions will convert the magnesium hydroxide layer Mg (OH) _2_ formed on the surface of the matrix into soluble MgCl_2_, thus damaging its protection to the matrix and accelerating the corrosion of Mg alloys.

In general, the reaction of Mg and its alloys in body fluid is as follows [43,44,45,46,47]:Anodic reaction: Mg → Mg^2+^ + 2e^−^(1)
Cathodic reaction: 2H_2_O + 2e^−^ → H_2_ + 2OH^−^(2)
Cathodic reaction: 2H_2_O + O_2_ + 4e^−^ → 4OH^−^(3)
Product formation: Mg^2 +^ +2OH^−^ → Mg(OH)_2_(4)

The potential interactions are shown in Figure 2.

### 3.2. Morphological Analysis of Mg_30_Zn_30_Sn_30_Sr_5_Bi_5_ Alloy before Corrosion

The SEM images of Mg_30_Zn_30_Sn_30_Sr_5_Bi_5_ are shown in Figure 3. The alloy is a multi-principal alloy, and the multiple numbers of elements also lead to the complex and variable phases that may appear in the smelting process. The following analysis can be made from the SEM image. Figure 3d,f show that Sn and Bi in the alloy do not overlap at all, and the amount of Sn is zero at the location where Bi occurs. This is completely consistent with the situation shown in the phase diagram of the Bi-Sn binary alloy, where there is no eutectic phase. As the solubility of Sn in Mg is limited, Sn will precipitate in the form of a simple substance when the temperature decreases. Therefore, it can be seen in Figure 3d that the position with the highest red brightness can be inferred as a simple substance Sn, exhibiting a large number of irregular blocky structures. Figure 3b,d show that Mg and Sn are most widely distributed in the alloy, and their overlapped area is also the largest. It can be perceived that Sn also appears where Mg occurs. Figure 3c shows that the content of Zn in the alloy is very low, and the atoms of the alloy elements are as much as the set amount of Mg, Zn, and Sn. There is almost no overlap between the highlighted yellow part in the figure and other figures, so it can be identified as a single substance Zn. Except for the highlighted part, other relatively dim positions basically belong to the coexistence of Sn, Sr, Mg, and Zn, the synthesis of multiple phases. In addition, in Figure 3a, there are several large and flat phase areas, which are mainly composed of Mg, Bi, and Sr.

The EDS image of Mg_30_Zn_30_Sn_30_Sr_5_Bi_5_ is shown in Figure 4. The following analysis can be made from the EDS image. After the preparation of the alloy, the loss of Zn is the largest, followed by Mg, and the loss of Sn is the smallest.

### 3.3. Phase Analysis of Mg_30_Zn_30_Sn_30_Sr_5_Bi_5_ Alloy

The XRD curve of Mg_30_Zn_30_Sn_30_Sr_5_Bi_5_ alloy is shown in Figure 5. Figure 6 shows the calculated Mole% phase curve (Figure 6a) and heat capacity–temperature curve (Figure 6b). Based on the analysis in 3.2, the following inference can be made. When the temperature is higher than 1004.72 °C, the alloy is completely liquid. At this temperature, Mg_3_Bi_2_ starts to precipitate, and the precipitation amount slowly increases, reaching saturation after the molar concentration reaches 12.08%. The mg_2_Sn phase starts to precipitate at 420.57 °C, and the MgZn_2_ phase at 333.26 °C. With the decrease in temperature, solid–liquid conversion occurs at 232.43 °C. In addition, the peritectic reaction of alloy occurs where the liquid phase + solid phase coexist between 230 °C and 250 °C. The molar percentage of MgZn_2_ decreases significantly at the same temperature range, while the molar percentage of Mg_2_Sn increases significantly. At this time, Sn displaces the Zn atoms in the MgZn_2_ phase. When Zn (1) atoms are replaced, a stable MgZn_2_ solid solution structure will be formed, while the MgZn_2_ solid solution structure formed by replacing Zn (1) and Zn (2) atoms simultaneously is unstable. Therefore, the molar concentration of the MgZn_2_ phase in the alloy increases with the further decrease in temperature, while the molar concentration of the Mg_2_Sn phase decreases. When the temperature reaches 90 °C, the molar concentration of Mg_2_Sn decreases obviously. At this time, because the solubility of Sn in the alloy decreases, it starts to precipitate in the form of a simple Sn phase. As shown in Figure 4, the results of XRD are highly similar to the calculated results. Based on the analysis of SEM and EDS, the overlapping position of Mg, Sn, and Bi in the alloy can be inferred as the coexistence area of the Mg_2_Bi_3_ (*The International Centre for Diffraction Data (ICDD^®^) 03-065-8732*) and Bi_3_Sr_5_ (*The International Centre for Diffraction Data (ICDD^®^) 03-065-2042*) phases. The area with a high overlap of Sn and Mg is the Mg2Sn (*The International Centre for Diffraction Data (ICDD^®^) 00-006-0190*) phase. There is also a large amount of simple substance Sn (*The International Centre for Diffraction Data (ICDD^®^) 03-065-7657*) and a small amount of simple substance Zn. In the coexistence area of Mg, Zn, Sn, and Sr, the content of Sn and Sr is generally very high, which demonstrates the refining effect of Sr on the Mg_2_Sn (*The International Centre for Diffraction Data (ICDD^®^) 00-006-0190*) phase and MgZn (*The International Centre for Diffraction Data (ICDD^®^) 00-001-1199*) phase.

Based on the analysis results of Section 3.2 and Section 3.3, the analysis of the complex phase of the alloy can be shown in Figure 7.

### 3.4. Electrochemical Corrosion Performance of Mg_30_Zn_30_Sn_30_Sr_5_Bi_5_ Alloy

#### 3.4.1. Potentiodynamic Polarization Result Analysis

The potentiodynamic polarization curve of Mg_30_Zn_30_Sn_30_Sr_5_Bi_5_ alloy and pure Mg is shown in Figure 8. The data obtained by polarization curve fitting are as follows: the self-corrosion potential of Mg_30_Zn_30_Sn_30_Sr_5_Bi_5_ alloy is −634.7 mV; its self-corrosion current density is 5.669 μA/cm^−2^; and the corrosion degradation rate is 0.066 mm/y. Meanwhile, the self-corrosion potential of pure Mg is −845.6 mV; its self-corrosion current density is 27.28 μA/cm^−2^; and the corrosion degradation rate is 0.32 mm/y. The test results show that the corrosion rate of the alloy is much lower than that of pure magnesium. All results are shown in Table 5.

According to the characteristics of alloy composition, the anodic polarization curve is related to the dissolution of the alloy. With the formation of Mg^2+^, the cathodic polarization curve is often related to the hydrogen evolution reaction. In the polarization curve, the self-corrosion potential of the alloy (−634.7 mV) is higher than that of pure Mg (−845.6 mV), which means that the corrosion resistance is significantly improved. Because the composition of the alloy is very complex, its reaction in the electrochemical corrosion experiment will also be very complicated. On the basis of the existing research results, the following analysis can be made. When the self-corrosion current density is low, there are fewer high-potential second phases in the alloy. The probability of galvanic corrosion, especially the one with Mg with low potential, is relatively small. The corrosion of the alloy surface can be reckoned as a slow and uniform corrosion. At this time, the anodic degradation is in a relatively stable state. However, with the increase in self-corrosion current density, the protective film on the surface is damaged, and its degradation rate increases obviously. It can be seen from the polarization curve that the corrosion resistance of the alloy is better than that of pure Mg in the slow corrosion stage. However, under the same condition, the corrosion rates of the alloy and pure Mg both increase significantly, and a partial overlap appears. At this point, their corrosion rates are basically the same. When the corrosion current density continues to increase, the corrosion resistance of the alloy once again presents a superior state to that of pure Mg. Therefore, the alloy has better corrosion resistance than pure Mg in anodic corrosion. Yet, in the hydrogen evolution corrosion of cathode, the alloy goes through a stage where the corrosion rate is higher than that of pure Mg, and later the two are basically the same. It can be understood that the increase in the high-potential second phase leads to serious galvanic corrosion, damages the protective film on the surface of the material, and thus causes corrosion to extend to the interior of the matrix. Sn, Zn, and Sr in the alloy can inhibit the hydrogen evolution reaction of the alloy cathode and mitigate the impact of cathodic corrosion to a certain extent [49,50,51].

#### 3.4.2. EIS Result Analysis

The Nyquist diagram of Mg_30_Zn_30_Sn_30_Sr_5_Bi_5_ alloy and pure Mg are shown in Figure 9. The impedance spectra of Mg_30_Zn_30_Sn_30_Sr_5_Bi_5_ alloy and pure Mg have similar characteristics, indicating that the alloy has the same corrosion mechanism as pure Mg. The curve radius of the alloy in the figure is much larger than that of pure Mg, and the corrosion resistance of the alloy is much higher than that of pure Mg.

To sum up, in the process of electrochemical corrosion experiment, the anode of Mg_30_Zn_30_Sn_30_Sr_5_Bi_5_ alloy shows superior corrosion resistance, while the cathode shows poor corrosion resistance due to more high potential second phases in the alloy. However, when the self-corrosion current density is low, its advantages in corrosion resistance are obvious, while the biological materials are mainly static immersion corrosion mode, so its corrosion resistance will be more prominent in biodegradation.

The electrochemical corrosion behavior in this study is characterized by EIS technology. The corresponding Nyquist diagram and Bode plot are shown in Figure 10, and the equivalent circuit for fitting the EIS curve is shown in Figure 10d. Corrosion film resistance Rct is parallel with constant phase element CPE1 describes the capacitance of corrosion film capacity in series with solution resistance Rs. From the existing experimental test results, the system shows the effect of the adsorption–corrosion inhibitor system, and its impedance diagram has only one time constant, showing a deformed single capacitive arc. Hydroxide film and oxide film form a protective film on the substrate surface, which makes the dispersion effect increase and the electric double-layer capacitance decrease. It can be seen from the Nyquist diagram that the corrosion resistance of the alloy is much higher than that of pure magnesium in the middle and high-frequency region (Figure 10a). The EIS fitting parameters are listed in Table 6.

### 3.5. Morphological Analysis of Mg_30_Zn_30_Sn_30_Sr_5_Bi_5_ Alloy after Corrosion

Figure 11 portrays the SEM image of Mg_30_Zn_30_Sn_30_Sr_5_Bi_5_ alloy after corrosion, Figure 12 portrays the EDS image of Mg_30_Zn_30_Sn_30_Sr_5_Bi_5_ alloy after corrosion, and the detailed morphology is shown in Figure 13. Figure 14 exhibits the XRD results. Based on the above analysis, when the electrochemical corrosion test was completed, an oxide film composed of various phases was formed on the material surface. Under its protection, the corrosion rate of the material surface decreased significantly. Mg_30_Zn_30_Sn_30_Sr_5_Bi_5_ alloy contains active metal ions with low electrode potential, such as Mg^2+^, which are prone to corrosion reaction in the complex alkaline physiological environment and react with OH^−^, PO_4_^2−^, HPO_4_^2−^, SO_4_^2−^ to form an insoluble hydrogen oxide film, oxide film, hydroxyapatite, etc., effectively preventing further corrosion of the alloy. Figure 11g and Figure 12, combined with Figure 11a, show that the content of the oxygen element is much higher than that of the non-corroded alloy. A large number of oxide phases appear in the alloy, such as Mg_4_O_4_ (*The International Centre for Diffraction Data (ICDD^®^) 01-087-0653*), NaO_3_ (*The International Centre for Diffraction Data (ICDD^®^) 01-070-5284*), MgO(*The International Centre for Diffraction Data (ICDD^®^) 01-071-1176*), Bi_7.38_Na_0.62_O_11.38_ (*The International Centre for Diffraction Data (ICDD^®^) 00-050-0371*), SrSnO_3_ (*The International Centre for Diffraction Data (ICDD^®^) 01-070-4389*), etc. These mixed phases form a protective film on the material surface, as is shown in Figure 13a (yellow highlight area just as “a”), where the mixture of surface hydroxide, oxide, etc., is evenly distributed and displays a dense granular morphology. With the deepening of corrosion, some Cl^−^ will replace (OH)^−^ and generate chlorides, which will damage the protective layer of the hydrogen oxide film. The formation of a total number of secondary phases leads to the formation of a galvanic couple, intensifies corrosion, and causes cracks on the surface of the material, as shown in Figure 13b (yellow highlight area just as “b”). In addition, the surface around the cracks is generally flat, and tends to be the corrosion acceleration caused by the Mg_2_Sn secondary phase.

As shown in Figure 13b, after the analysis of the morphology of the corroded alloy surface, the corroded surface was removed, and an XRD test was conducted on the alloy substrate. It can be confirmed that after the SBF electrochemical corrosion test, the alloy phase has changed to a certain extent, including the appearance of Sr_4_Zn_44_ (*The International Centre for Diffraction Data (ICDD^®^) 03-065-5701*), Bi_3_Sr_5_ (*The International Centre for Diffraction Data (ICDD^®^) 00-031-0198*), Mg_23_Sr_6_ (*The International Centre for Diffraction Data (ICDD^®^) 03-065-1777*), etc. The loss of Sn (*The International Centre for Diffraction Data (ICDD^®^) 00-004-0673*) is the largest, and the loss of Mg, Bi, and Sr is less under the protection of various hydrogen oxide films and oxide films. Zn, as an important element in the alloy, has suffered serious loss during the preparation process, which is more serious after the SBF electrochemical corrosion experiment. This phenomenon deserves further research.

## 4. Conclusions

The main results of this study include the following aspects:
Based on the theory of biomaterials and the high entropy alloy theory, the elements of the alloy were determined to be Mg, Zn, Sn, Sr, Bi, and the atomic percentage of the element was 30: 30: 30: 5: 5. High-purity metal elements was used for smelting to produce the alloy through the magnetic levitation smelting technology. The initial purpose of this study was to find ways to improve the corrosion resistance of Mg alloy biomaterials. Therefore, this study mainly demonstrated its feasibility by comparing the electrochemical corrosion resistance of alloys and pure Mg.The corrosion rate of Mg_30_Zn_30_Sn_30_Sr_5_Bi_5_ alloy is 0.066 mm/y, which is much lower than that of pure magnesium (0.32 mm/y) under the same test conditions. Combined with the existing research results, the corrosion rate of common magnesium alloys is higher than that of the Mg_30_Zn_30_Sn_30_Sr_5_Bi_5_ alloys prepared in this study. Such as, the corrosion rate of ZK60 is 2.6791 mm/y, AM60 is 1.9573 mm/y, AZ31 is 3.1404 mm/y, etc. [52,53].The electrochemical corrosion experiment demonstrates that the self-corrosion current density of Mg_30_Zn_30_Sn_30_Sr_5_Bi_5_ alloy was 2.52 μA/cm^−2^, and that of pure magnesium was 12.39 μA/cm^−2^. The self-corrosion potential of the alloy is −634.7 mV, while that of pure magnesium was −845.6 mV. The corrosion degradation rate is 0.066 mm/y, and that of pure magnesium is 0.32 mm/y.Based on analysis of the SEM, EDS, and XRD test results of Mg_30_Zn_30_Sn_30_Sr_5_Bi_5_ alloy before and after the corrosion, it could be inferred that there were many different phases in the alloy, and various elements had a certain impact on the corrosion resistance of the material. As the important elements, Sn and Zn exist in the alloy in the form of the Sn phase, Mg_2_Sn phase, MgZn_2_, etc., which hinder the diffusion of Cl^−^ ions in m-SBF and protect the Mg (OH) phase generated by Mg^2+^ ions. Meanwhile, the formation of Mg_4_O_4_, NaO_3_, MgO, Bi_7.38_Na_0.62_O_11.38_, SrSnO_3_, and other oxides also served as a protective layer on the alloy surface to prevent further corrosion. The presence of hydroxyapatite also played a role in protecting the surface of the alloy by slowing down the occurrence of galvanic corrosion, and thus met the design requirements for a low corrosion rate. After corrosion, the alloy surface displayed two main states. One is the presence of a mixture of hydroxide, oxide, hydroxyapatite, and other substances in dense granular form on the material surface. The other is cracks on the alloy surface due to the intensification of corrosion, because the protective layer was damaged by the increase in Cl^−^ and higher secondary phases and later the formation of MgCl and other phases.The research results proved that the alloy prepared in this study displayed good corrosion resistance at low self-corrosion current density in the electrochemical corrosion test in the m-SBF. Anodic corrosion always shows good corrosion resistance, but cathodic corrosion shows poor corrosion resistance due to the increase in self-corrosion current density.

This study mainly conducted the electrochemical corrosion analysis on the alloy material. As an in vitro simulation experiment of biomaterials, it is an important part of material performance research. Therefore, these experimental results can support the further exploration of the material. Relevant research needs to be carried out in the future to ensure the integrity of the literature, and also to enable the results to obtain more abundant data support. In addition, as the main element of the alloy in this study, Zn has suffered huge losses both in the preparation process and the electrochemical corrosion test experiment. In the following studies, the cause should be determined through further testing to adjust the content of Zn in the alloy. There are various potentials for the proportion of elements in the alloy. A more reasonable proportion of elements in the alloy can be achieved in future studies to obtain better biomaterial performance and corrosion resistance.

## 5. Patents

A patent for invention was granted. Invention name: Mg-based biomaterials based on high entropy alloy theory and their preparation methods and applications. Patent Application No.: ZL 2021 1 1500334.1.

## Figures and Tables

**Figure 1 materials-16-01939-f001:**
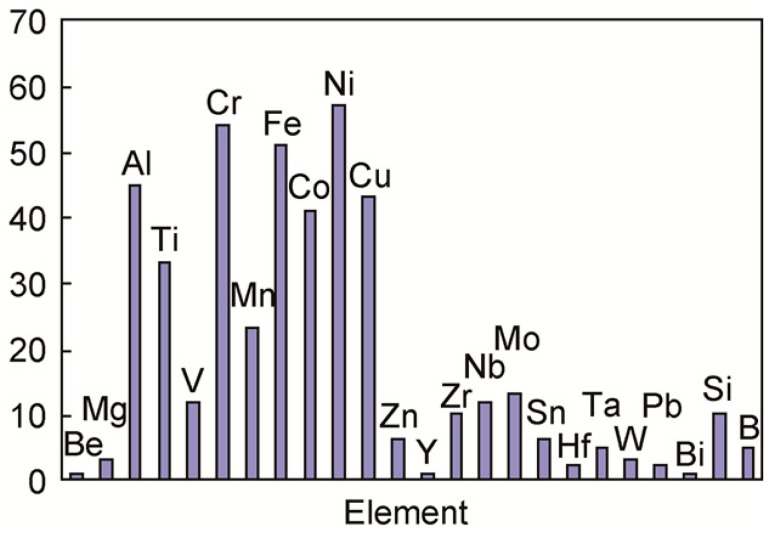
Strip chart of common elements in high entropy alloy research (the vertical coordinate of the picture is the statistics data of various elements from different kinds of HEAs).

**Figure 2 materials-16-01939-f002:**
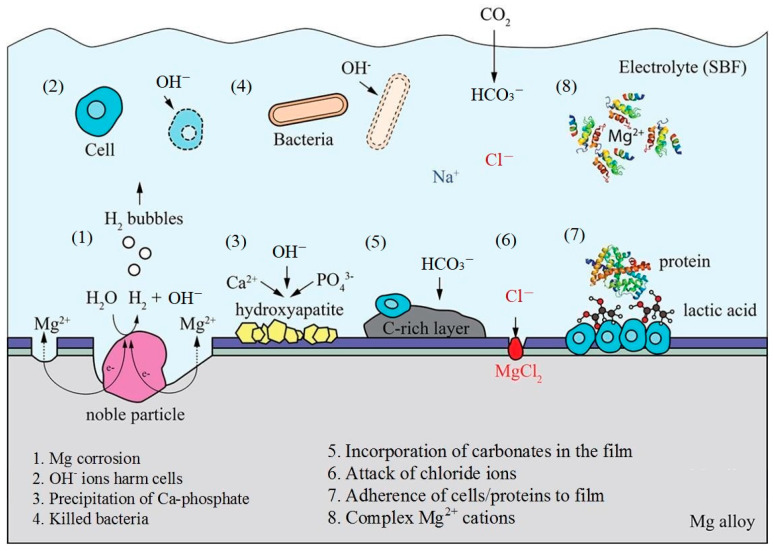
Potential interactions between corroded magnesium alloy surface and physiological environment [48].

**Figure 3 materials-16-01939-f003:**
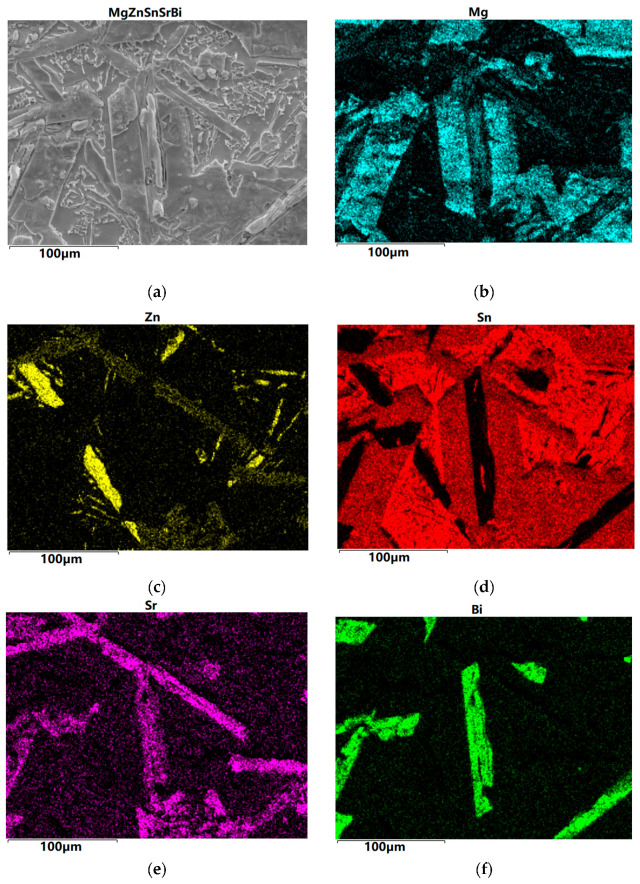
SEM image of Mg_30_Zn_30_Sn_30_Sr_5_Bi_5_ at% alloy. (**a**) SEM images observed at the cross section of Mg30Zn30Sn30Sr5Bi5 at% alloy; (**b**) The distribution of individual elements Mg; (**c**) The distribution of individual elements Zn; (**d**) The distribution of individual elements Sn; (**e**) The distribution of individual elements Sr; (**f**) The distribution of individual elements Bi.

**Figure 4 materials-16-01939-f004:**
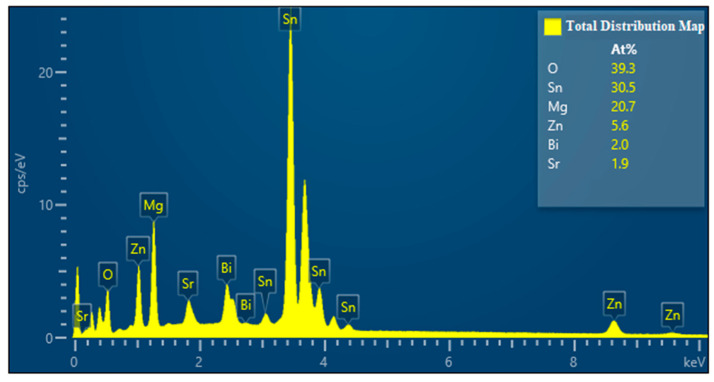
EDS image of Mg_30_Zn_30_Sn_30_Sr_5_Bi_5_ alloy.

**Figure 5 materials-16-01939-f005:**
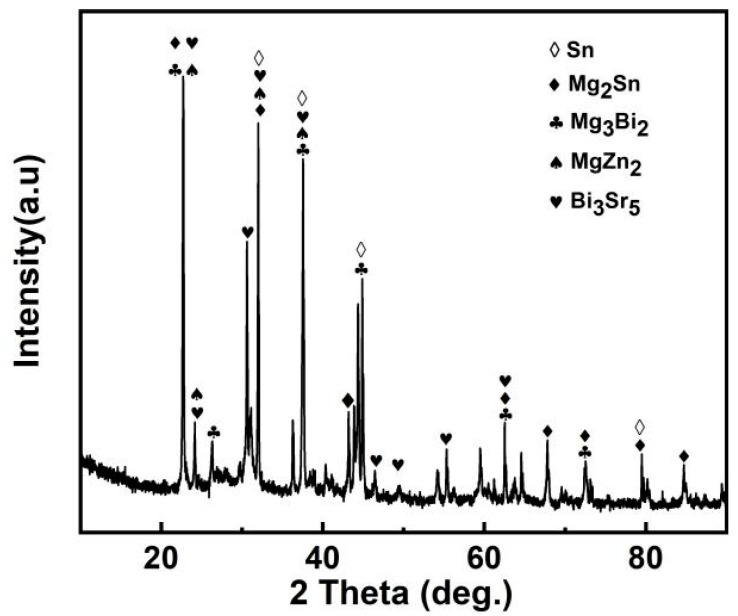
XRD of Mg_30_Zn_30_Sn_30_Sr_5_Bi_5_ alloy.

**Figure 6 materials-16-01939-f006:**
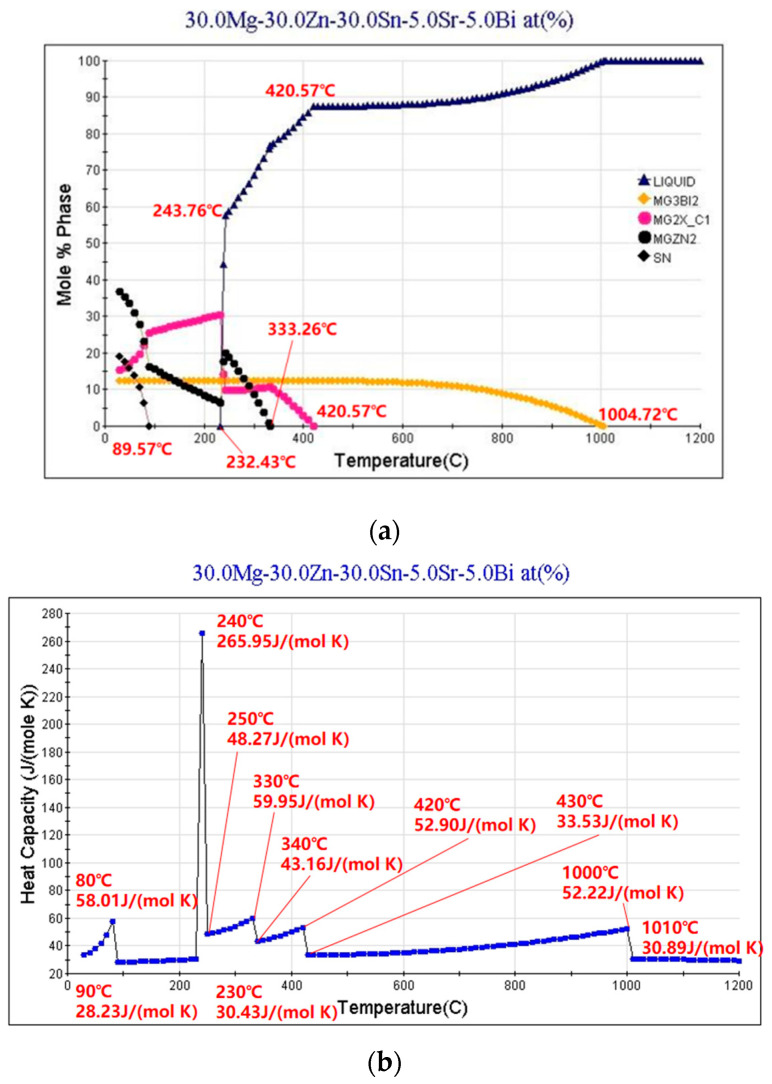
Mole % phase curve and heat capacity–temperature curve of Mg_30_Zn_30_Sn_30_Sr_5_Bi_5_ alloy ((**a**) Mole % phase curve. (**b**) Heat capacity–temperature curve).

**Figure 7 materials-16-01939-f007:**
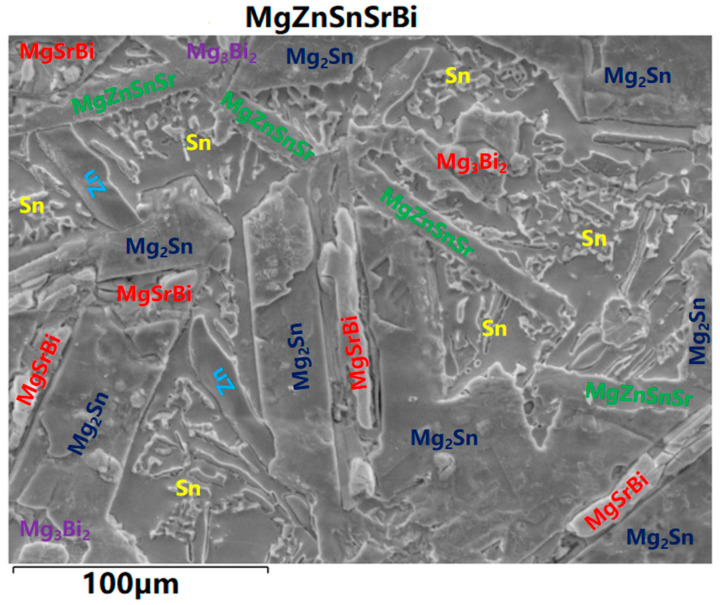
Analysis result image of phase.

**Figure 8 materials-16-01939-f008:**
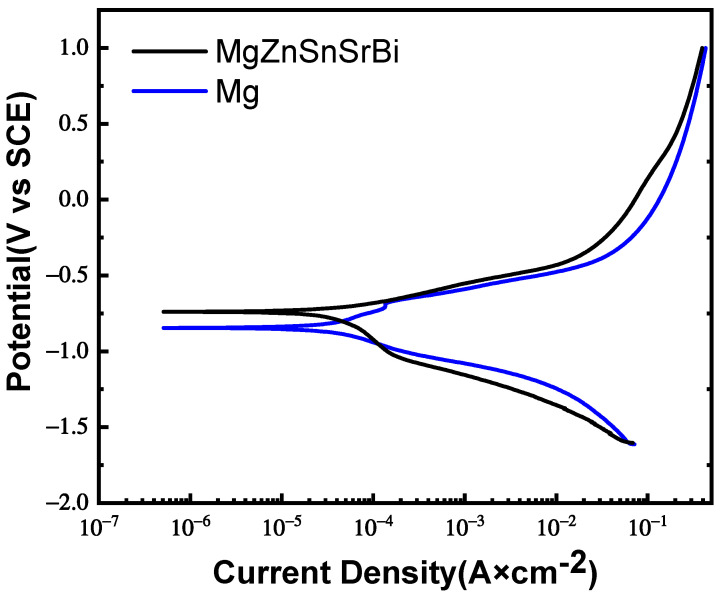
Polarization curve of Mg_30_Zn_30_Sn_30_Sr_5_Bi_5_ alloy and pure Mg.

**Figure 9 materials-16-01939-f009:**
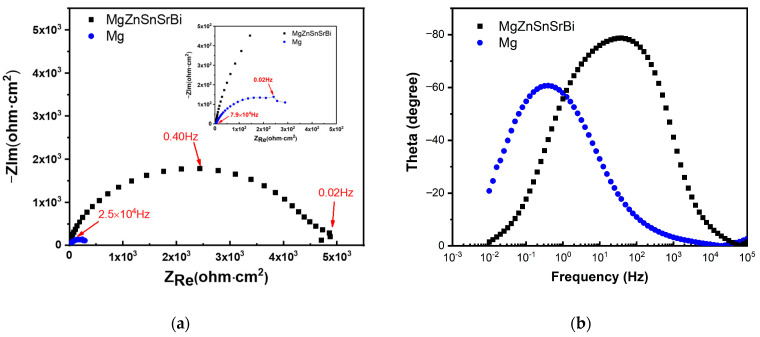
Nyquist and Bode curve of Mg_30_Zn_30_Sn_30_Sr_5_Bi_5_ alloy and pure Mg. (**a**) Nyquist curve of Mg30Zn30Sn30Sr5Bi5 alloy and pure Mg; (**b**) Bode curve of Mg_30_Zn_30_Sn_30_Sr_5_Bi_5_ alloy and pure Mg).

**Figure 10 materials-16-01939-f010:**
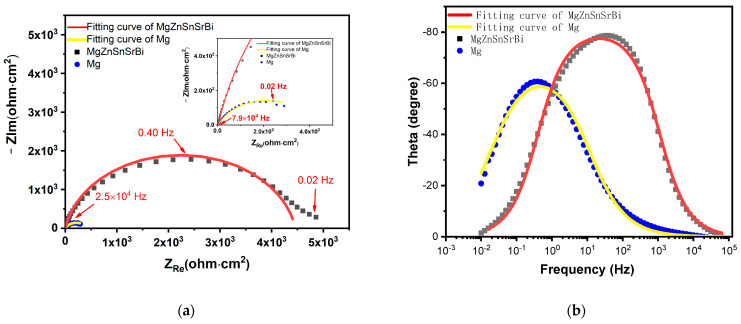
Corrosion behavior of the samples evaluated by electrochemical corrosion test ((**a**) Nyquist curves, (**b**) Bode plot–phase angle curve, (**c**) Bode plot–impedance module, and (**d**) equivalent circuit used for fitting EIS data).

**Figure 11 materials-16-01939-f011:**
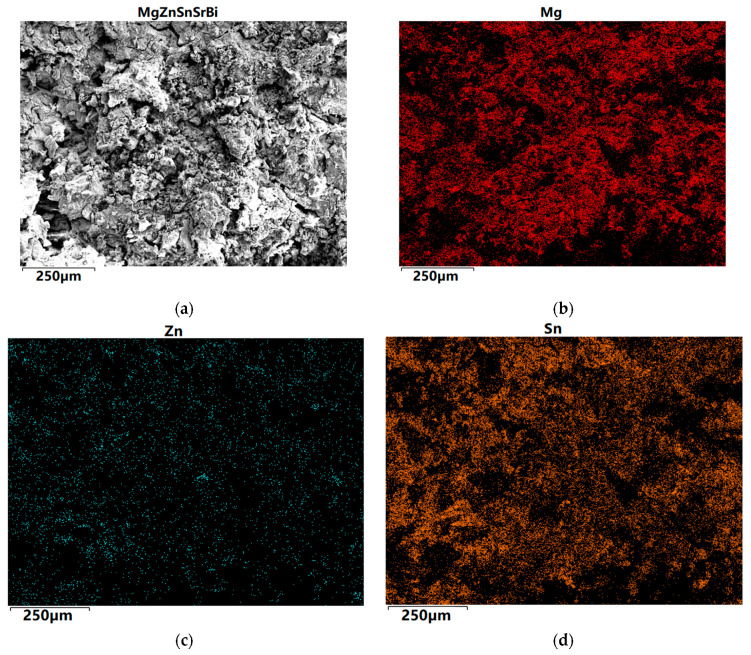
SEM image of Mg_30_Zn_30_Sn_30_Sr_5_Bi_5_ alloy. (**a**) SEM images observed at the cross section of Mg30Zn30Sn30Sr5Bi5 at% alloy; (**b**) The distribution of individual elements Mg; (**c**) The distribution of individual elements Zn; (**d**) The distribution of individual elements Sn; (**e**) The distribution of individual elements Sr; (**f**) The distribution of individual elements Bi; (**g**) The distribution of individual elements O.

**Figure 12 materials-16-01939-f012:**
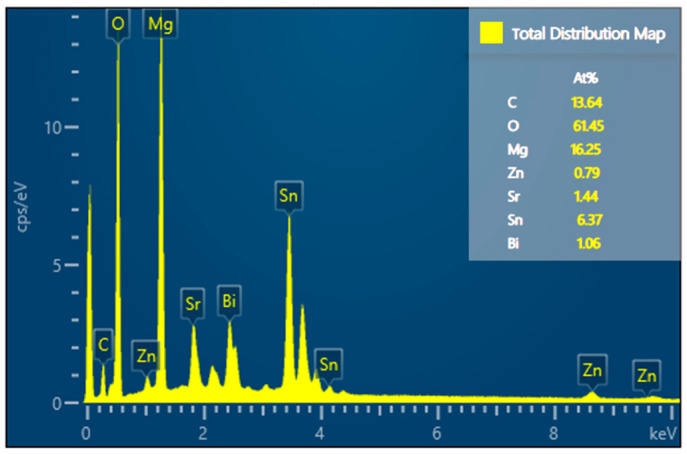
EDS image of Mg_30_Zn_30_Sn_30_Sr_5_Bi_5_ alloy.

**Figure 13 materials-16-01939-f013:**
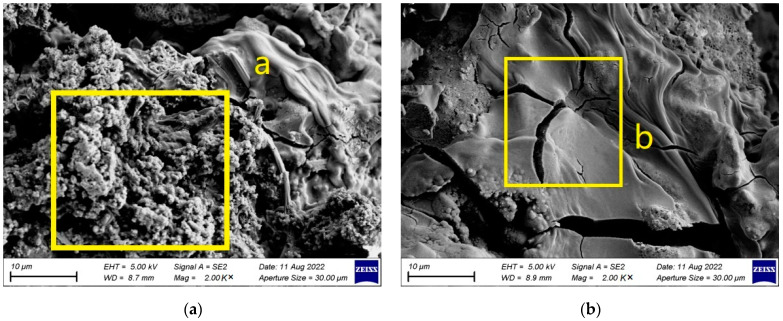
The local amplification map of morphology after corrosion of Mg_30_Zn_30_Sn_30_Sr_5_Bi_5_ alloy. (**a**) Position a displays a dense granular morphology which is mixtured of surface hydroxide, oxide, etc.; (**b**) Position b displays cracks on the surface of the material by intensifies corrosion.

**Figure 14 materials-16-01939-f014:**
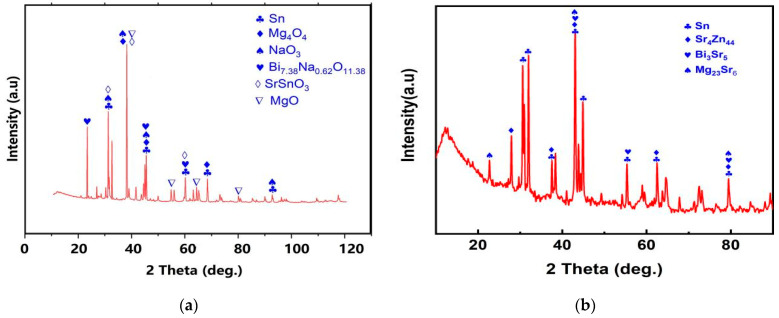
XRD diagram of Mg_30_Zn_30_Sn_30_Sr_5_Bi_5_ alloy. (**a**) XRD diagram of Mg_30_Zn_30_Sn_30_Sr_5_Bi_5_ alloy surface which has been corrosioned; (**b**) XRD diagram of Mg_30_Zn_30_Sn_30_Sr_5_Bi_5_ alloy surface after removing corrosion Layerx.

**Table 1 materials-16-01939-t001:** The selection basis of alloying elements.

Criteria and Principles	Content
High entropy alloy theory	The material contains five or more major elements [20];
	The atomic percentage of each element is between 5% and 35% [21];
	In order to easily form solid solution phase, the difference in atomic size of each element should be less than or equal to 7% [22];
	The elements that are easier to form high entropy alloy with, summarized by Scholars in the field of high entropy alloy research, are shown in Figure 1 [23].
Element implantability requirements [24,25]	Metal elements must have good biological characteristics and are safe for organisms without any toxic side effects;
	One of the main goals of medical Mg and Mg alloy design is to improve the method, which can realize solution strengthening, fine grain strengthening, and aging strengthening by adding different elements;
	The metal elements should be the ones that appear with high frequency currently in the research of Mg alloy biomaterials.

**Table 2 materials-16-01939-t002:** Summary of the metal elements.

No.	Element Types	Atomic Radius nm	Biological Characteristics
1	Mg	136	Mg and Mg alloys have good biocompatibility. Mg is an important element involved in human metabolism and maintenance of the normal operation of human tissues. According to the new RDA standard of the United States, the mass of Mg in normal adults is 21–28 g, and the daily intake of adults is 180–350 mg. A total of 53% of Mg exists in bones, and the rest is usually in muscle tissues and organs, such as liver, brain, and kidney. In addition, Mg is an activator of many enzymes, a common regulator of protein synthesis and muscle contraction, and a stabilizer of DNA and RNA [26,27].
2	Zn	125	Zn is one of the essential trace elements for human body, and plays an extremely important role in human growth, immunity, endocrine, and other physiological processes. Therefore, it is called “the flower of life” and “the source of intelligence”. Zn can contribute to the normal function of many enzymes; promote wound healing; improve nerve transmission and synapse formation; support protein and DNA synthesis as well as the sense of taste and smell; enhance immunological activity. Zn deficiency leads to delayed responses to T cell-dependent and T cell-independent antigens [28,29,30].
3	Sn	140	Sn is a trace element in the human body, which is relatively non-toxic within a certain range. In the human body, Sn can not only improve the activity of a variety of enzymes, but also interfere with the metabolism of Zn, Cu, and Ca, changing their concentrations in human tissues. In the existing Mg-Sn alloy studies, researchers have confirmed that an appropriate amount of Sn in the human body has good biocompatibility and blood compatibility, and will not cause cytotoxicity [31,32,33].
4	Sr	191	Sr is a natural bone-seeking element that accumulates in bones due to its close physical and chemical properties to Ca. It can reduce bone absorption; stimulate the growth of osteoblasts; and enhance bone strength and bone mineral density. In addition, the degradation of Mg-Sr alloy is conducive to the deposition of hydroxyapatite and bone mineralization [34,35,36,37].
5	Bi	146	Bi is not an essential element for animals and plants, and the trace amount is harmless to human body. It is often used in medicine in the form of compounds, such as contrast agents, bismuth potassium tartrate, etc. Although Bi is similar to Pb, it is harmless to human body and is a “green metal” because there has been a trend for Bi to replace Pb as a green, environmentally friendly material. Adding Bi to Mg alloys can promote bone formation without generating gas, and has great potential clinical application value [24].

**Table 3 materials-16-01939-t003:** The particle size and purity of the raw materials.

No.	Name	Purity	Size	Shape
1	High-purity Mg	99.99 wt.%	Ø4 × 4 mm	Particles
2	High-purity Zn	99.999 wt.%	1–3 mm	Particles
3	High-purity Sn	99.999 wt.%	1–6 mm	Particles
4	High-purity Sr	99.9 wt.%	1–3 cm	Particles
5	High-purity Bi	99.999 wt.%	1–3 mm	Particles (spherical approximately)

Notes: The supplier for each raw material is Beijing Yijin New Material Technology Co., Ltd. https://yijinxc.cn.china.cn, accessed on 28 June 2022).

**Table 4 materials-16-01939-t004:** Ionic concentrations (mM) of blood plasma and proposed SBF formulations (pH 7.4–7.5 at 36.5 °C).

Formulation	Na^+^	K^+^	Mg^2+^	Ca^2+^	Cl^−^	HCO^−3^	HPO_2_^−4^	SO_2_^−4^	pH
Blood Plasma	142.0	5.0	1.5	2.5	103.0	27.0	1.0	0.5	7.4–7.5
m-SBF	142.0	5.0	1.5	2.5	103.0	10.0	1.0	0.5	7.4
SBF	142.0	5.0	1.5	2.5	147.8	4.2	1.0	0.5	7.4

**Table 5 materials-16-01939-t005:** Potentiodynamic polarization result.

Material	E_corr_	i_corr_	Rp	Corrosion Rate
Mg_30_Zn_30_Sn_30_Sr_5_Bi_5_ alloy	−634.7	5.669	4.708 × 10^3^	2.591
Pure Mg	−845.6	27.28	3.093 × 10^2^	12.7

Units: Ecorr: mV, icorr: μA/cm^−2^, Rp: Ω·cm^2^, corrosion rate: mpy.

**Table 6 materials-16-01939-t006:** Equivalent circuit fitting results of EIS data.

Samples	Rs (Ω cm^2^)	CPE-T	CPE-P	Rct (Ω cm^2^)
Mg_30_Zn_30_Sn_30_Sr_5_Bi_5_	3.666	9.7436 × 10^−5^	0.89675	4454
Pure Mg	3.348	0.011928	0.75321	420.9

## Data Availability

Sharing and archiving research data.

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
