# Peer review of "Study on Material Design and Corrosion Resistance Based on Multi-Principal Component Alloying Theory"

_materials, 2023, doi:10.3390/ma16051939_

Round 1

Reviewer 1 Report

The main question addressed by the research is the stability of magnesium materials in body fluids by preparing a new biomaterial Mg30Zn30Sn30Sr5Bi5. This topic is entirely relevant in the field of materials and nanocomposites. It addresses the importance of the use  of vacuum magnetic levitation melting technology  in order to prepare new biomaterials. Compared with other published material, the magnesium alloy is several times more resistant against corrosion at lower density currents. the authors should try to find if there is any other combination of metals that can increase the stability of magnesium materials in the body 's environment.

Remarks

P1, line 31: Put Nyquist instead of Nquist

P6, 158: Put the unity of pressure mbar or other in the sentence when the vacuum reaches 10-3

P6, 195: Put degradation instead of Dedgradation

P7, 201: Put degradation instead of degrada-tion

P7, 209: Put Figure 2 instead of Figure 3.

P8, 217: Put Figure 4 instead of Figure 5

P11, 318: Put Bode plot instead of Porter diagram

Author Response

Dear Reviewer,

  We appreciate you very much for your positive and constructive comments and suggestions on out manuscripts entitled “ Study on Material Design and Corrosion Resistance Based on Multi-principal Component Alloying Theory “ (Manuscript ID: materials-2196022). Those comments are all valuable and very helpful for revising and improving our paper, as well as the important guiding significance to our researches.

The main question addressed by the research is the stability of magnesium materials in body fluids by preparing a new biomaterial Mg30Zn30Sn30Sr5Bi5. This topic is entirely relevant in the field of materials and nanocomposites. It addresses the importance of the use  of vacuum magnetic levitation melting technology  in order to prepare new biomaterials. Compared with other published material, the magnesium alloy is several times more resistant against corrosion at lower density currents. the authors should try to find if there is any other combination of metals that can increase the stability of magnesium materials in the body 's environment. 

Using "multi-principal component alloying" method to improve the corrosion resistance of degradable Mg alloy biomaterials is the main research direction of our team. At present, the main consideration is the feasibility of the method, and then we will adjust the atomic percentage of alloying elements to find the best ratio scheme. In the future, we will adjust the elements of the alloy, try to replace the elements in the alloy, and find more elements to combine.

P1, line 31: Put Nyquist instead of Nquist

The Nyquist diagram shows that the self-corrosion potential of the alloy is much higher than that of pure Mg. In general, under the condition of low self-corrosion current density, the alloy materials display an excellent corrosion resistance.

P6, 158: Put the unity of pressure mbar or other in the sentence when the vacuum reaches 10-3

Step 2. Vacuum the furnace. When the vacuum degree reaches 10-3MPa, clean the gas in the furnace with argon twice. After that, argon is filled into the whole furnace to ensure that all smelting and casting processes are conducted under argon protection.

P6, 195: Put degradation instead of Dedgradation

3.1 Corrosion and Degradation Mechanism of Mg and Mg alloys

P7, 201: Put degradation instead of degrada-tion

These elements will have various effects on the corrosion degradation process of Mg alloys.

P7, 209: Put Figure 2 instead of Figure 3.

The potential interactions are shown in Figure 2.

P8, 217: Put Figure 4 instead of Figure 5

The content of this part has been greatly adjusted and the serial number has changed. Figure 3 and Figure 4 instead Figure 3. Figure 5 instead Figure 5. Figure 5 instead Figure 6. Added Figure 7.

P11, 318: Put Bode plot instead of Porter diagram

 Bode plot instead of  Porter diagram 

Reviewer 2 Report

Review “Study on Material Design and Corrosion Resistance Based on 2 Multi-principal Component Alloying Theory”

Materials

1. Introduction

Line 42: “….and fatigue strength[2]”. Use a space between strength and the reference.

Line 46: What is meaning of “metal icons”?

Line 51: Space before the unit GPa

Lines 60-65: The information must be referenced.

Lines 104-107: This idea is confuse…. try to explain it clearly: “The reason why high entropy alloy theory is used to design the alloy is that compared with the saturated development of traditional alloys, the development of multibody alloys is expected to break the limitations of alloy material development.”.

Line 131: It must correct the table label “Table 1. This is a table. Tables should be placed in the main text near to the first time they are cited.”

Table 1. It seems that word “Figure” is missing: “… are shown in 1 [22].”

Line 152: It must correct the table label “Table 3. This is a table. Tables should be placed in the main text near to the first time they are cited.”

In table 3: In column Size: What is the meaning of “1-3mm Similar spherical”.

Line 158: What unit is used in “vacuum degree reaches 10-3”?

Lines 162-163: Use kW instead of KW.

Line 172: use pure instead Pure.

Line 177: Nyquist instead Nyquist

Line 188: is it correct the use of the word “composites” in this context?

Line 195: Correct typo “Dedgradation”

Line 201: Correct “degrada- tion”

Lines 214 and 218: It is no well use the expression "SEM diagram", instead of that, it is recommend use "SEM image"

Lines 217-218-Page 8. The authors indicate in the text: “It can be clearly seen in Figure 3(a), the SEM diagram, that the alloy contains many phases.” In figure 3, the authors must show which phases are present in the submitted SEM image, using some way to indicate them directly on the image (arrows, circles, etc.),

It is recommended that the SEM and EDS results be presented separately, so that they can be better analyzed.

Line 213: item 3.2. The point being about morphological analysis, it should be done more rigorously. That is, identify the phases present and mark them in order to distinguish them, which is considered important since it is a new alloy.

Lines 219-220: Please check the number of figure: “the following inferences can be made as is shown in Figure 6.”

Line 259: For more clarity, indicate the meaning of this nomenclature inside parenthesis “i.e. (ICDD 03-065-8732)” as The International Centre for Diffraction Data (ICDD®).

In general:

Check spaces before references, spaces between value and units.

Be careful with the use of capitals in some word that do not need it.

Is it possible to include other test results for chemical composition of the alloy?

Author Response

Dear Reviewer,

  We appreciate you very much for your positive and constructive comments and suggestions on out manuscripts entitled “ Study on Material Design and Corrosion Resistance Based on Multi-principal Component Alloying Theory “ (Manuscript ID: materials-2196022). Those comments are all valuable and very helpful for revising and improving our paper, as well as the important guiding significance to our researches.

  1. Introduction

Line 42: “….and fatigue strength[2]”. Use a space between strength and the reference.

All the spaces before references and the spaces between value and units in the article have been checked and modified.

Line 46: What is meaning of “metal icons”?

metal ions instead of metal icons.

Line 51: Space before the unit GPa

All the spaces before references and the spaces between value and units in the article have been checked and modified.

Lines 60-65: The information must be referenced.

Added a referenced, “6.Rongxiang Wang, Lixin Hong, Xiaobo Zhang. Research progress in corrosion resistance of biomedical magnesium alloys [J]. Journal of Materials Engineering, 2021, 49(12): 14-27.”

Lines 104-107: This idea is confuse…. try to explain it clearly: “The reason why high entropy alloy theory is used to design the alloy is that compared with the saturated development of traditional alloys, the development of multibody alloys is expected to break the limitations of alloy material development.”.

“This study attempts to develop a degradable Mg alloy biomaterial with better corrosion resistance by multi-principal alloying method. With the continuous development of material science, the literature of binary and ternary alloys has been well-established. But the research of multi-principal alloys provides a new possibility and is expected to break this limitation. High entropy alloy is a typical representative of multi-principal alloys.” instead The reason why high entropy alloy theory is used to design the alloy is that compared with the saturated development of traditional alloys, the development of multibody alloys is expected to break the limitations of alloy material development.

Line 131: It must correct the table label “Table 1. This is a table. Tables should be placed in the main text near to the first time they are cited.”

Amend it to " The selection basis of alloying elements".

Table 1. It seems that word “Figure” is missing: “… are shown in 1 [22].”

… are shown in Figure 1 [22] instead of … are shown in 1 [22].

Line 152: It must correct the table label “Table 3. This is a table. Tables should be placed in the main text near to the first time they are cited.”

Amend it to " The particle size and purity of the raw materials".

In table 3: In column Size: What is the meaning of “1-3mm Similar spherical”.

spherical approximately instead of similar spherical.

Line 158: What unit is used in “vacuum degree reaches 10-3”?

The unit is MPa.

Lines 162-163: Use kW instead of KW.

All of KW have been instead by kW.

Line 172: use pure instead Pure.

pure Mg instead of Pure Mg.

Line 177: Nyquist instead Nyquist

Nyquist and Bode plots instead of nyquist and bode plot.

Line 188: is it correct the use of the word “composites” in this context?

The word composites in this context is discorrect, has been removed.

Line 195: Correct typo “Dedgradation”

Amend Dedgradation to Degradation

Line 201: Correct “degrada- tion”

Amend degrada-tion to degradation.

Lines 214 and 218: It is no well use the expression "SEM diagram", instead of that, it is recommend use "SEM image"

The SEM image instead of The SEM and EDS diagram.

the SEM image instead of  the SEM diagram.

Lines 217-218-Page 8. The authors indicate in the text: “It can be clearly seen in Figure 3(a), the SEM diagram, that the alloy contains many phases.” In figure 3, the authors must show which phases are present in the submitted SEM image, using some way to indicate them directly on the image (arrows, circles, etc.),

Add new pictures to the paper.

It is recommended that the SEM and EDS results be presented separately, so that they can be better analyzed.

The SEM and EDS results are presented separately.

Line 213: item 3.2. The point being about morphological analysis, it should be done more rigorously. That is, identify the phases present and mark them in order to distinguish them, which is considered important since it is a new alloy.

The content of 3.2 has been rewritten.

Lines 219-220: Please check the number of figure: “the following inferences can be made as is shown in Figure 6.”

Figure 6 has been added.

Line 259: For more clarity, indicate the meaning of this nomenclature inside parenthesis “i.e. (ICDD 03-065-8732)” as The International Centre for Diffraction Data (ICDD®).

Amend all ICDD xx-xxx-xxxx to The International Centre for Diffraction Data (ICDD®) xx-xxx-xxxx.

In general:

Check spaces before references, spaces between value and units.

We checked the full text and added missing spaces.

Be careful with the use of capitals in some word that do not need it.

We reviewed the full text and corrected the capitals problem.

Is it possible to include other test results for chemical composition of the alloy?

We tested the material by TEM. Because there are still some theories and viewpoints that need to be further improved, they are not presented in this paper.

Reviewer 3 Report

The manuscript presents an interesting study of the corrosion behaviour of a magnesium alloy studied by EIS and potentiodynamic polarization in SBF. However, the paper needs major revisions before it is processed further, some comments follow:

Abstract:

The abstract must be improved. Please highlight the novelty of the study and make it shorter regarding the conclusions (present only the main conclusion);

 Materials and methods

Lines 127-130. Please split this sentence into many in order to be more understandable.

Line 131. Table 1 has no title.

Please provide the supplier for each raw material.

Line 152. Table 3 has no title.

Line 158. Introduce the furnace type/manufacturer.

Line 171. I consider that is unnecessary the information from the bracket, please remove it.

Subsection 2.3. Please introduce information about the equipment and the software used for data acquisition and processing.  Also, introduce the surface exposed and the type of auxiliary electrode.

What is the pH of the SBF used?

 Results

The discussion section is missing. The authors have 2 options: a-rename this section Results and discussion and improve the section by comparing the results with other studies or b- introduce a new section name Discussion.

Figure 2 is not clear. Please replaced it.

Line 214. SEM and EDS images are not diagrams. Please rewrite.

Lines 214-217. The authors are present in the text figure 3 and the next figure 5. The figure must be presented in the order which appears in the text.

Figure 3. Remove the (a), (b), (c)…. Before the name of the element from the figure. Also, in figure (a) please introduce figure labels in order to highlight the interest zones. Please increase the size of figure (g).

I recommend discussing the results of the XRD making reference to other studies.

How were determined the results from Figure 5? Please write in the Material and methods section.

Subsection 3.4. First of all, the potentiodynamic polarization results and EIS results must be discussed in different subsections.

Potentiodynamic polarization results: Please introduce a table with the values of current density, corrosion rate, corrosion potential and polarization resistance. And discuss all of them. How was calculated the corrosion rate? Regarding Figure 6, why did the authors put the Nyquist diagram next to the polarization curve?

Also, why the Nyquist diagram for Figure 7 it is different? And for each sample are the results from Figure 7? Please clearly specify.

The results of the corrosion test must be discussed by comparing the obtained results with others from the literature.

Subsection 3.5. Please improve the discussion and also add labels to highlight the interested zone in Figures 8(a) and 9.

Results and Limitation

This section must be replaced with the Conclusion section.

Author Response

Dear Reviewer,

  We appreciate you very much for your positive and constructive comments and suggestions on out manuscripts entitled “ Study on Material Design and Corrosion Resistance Based on Multi-principal Component Alloying Theory “ (Manuscript ID: materials-2196022). Those comments are all valuable and very helpful for revising and improving our paper, as well as the important guiding significance to our researches.

Abstract:

The abstract must be improved. Please highlight the novelty of the study and make it shorter regarding the conclusions (present only the main conclusion);

The abstract has been improved.

“Abstract: This study mainly attempts to develop Mg-based alloy materials with excellent corrosion resistance by means of multi-principal alloying. The alloy elements are determined based on the multi-principal alloy elements and the performance requirements of the components of biomaterials.  Mg30Zn30Sn30Sr5Bi5 alloy was successfully prepared by vacuum magnetic levitation melting. Through the electrochemical corrosion test with m-SBF solution (pH7.4) as the electrolyte, the corrosion rate of alloy Mg30Zn30Sn30Sr5Bi5 alloy decreased to 21% of pure Mg. It could also be seen from the polarization curve that when the self-corrosion current density is low, the alloy shows superior corrosion resistance. Nevertheless, with the increase of self corrosion current density, although the anodic corrosion performance of the alloy is obviously better than that of pure Mg, the cathode shows the opposite situation. The Nyquist diagram shows that the self-corrosion potential of the alloy is much higher than that of pure Mg. In general, under the condition of low self-corrosion current density, the alloy materials display an excellent corrosion resistance. It is proved that the multi-principal alloying method is of positive significance for improving the corrosion resistance of Mg alloys.”

Materials and methods

Lines 127-130. Please split this sentence into many in order to be more understandable.

The sentence was rewritten. As a new type of Mg alloy biomaterial, the material A for corrosion performance test in this study is a multi-component alloy material. The element proportion standard is based on the Multi-principal element alloying theory and the element selection is on the basis of implantability of organisms.

Line 131. Table 1 has no title.

Amend it to " The selection basis of alloying elements".

Please provide the supplier for each raw material.

The supplier for each raw material is Beijing Yijin New Material Technology Co., Ltd. https://yijinxc.cn.china.cn

Line 152. Table 3 has no title.

Amend it to " The particle size and purity of the raw materials".

Line 158. Introduce the furnace type/manufacturer.

Vacuum electromagnetic levitation melting operation is completed by Beijing Yijin New Material Technology Co., Ltd. (https://yijinxc.cn.china.cn). The manufacturer of vacuum induction electromagnetic levitation smelting furnace is Jinzhou Taihe District Weili Metallurgical Equipment Factory (http://www.jzwlyj.cn/index.php/cn/Index/index.html), Equipment model: ZG2- XF, Capacity:2 kg, Mg30Zn30Sn30Sr5Bi5 alloy weight: 1 kg.

Line 171. I consider that is unnecessary the information from the bracket, please remove it.

It has been removed.

Subsection 2.3. Please introduce information about the equipment and the software used for data acquisition and processing.  Also, introduce the surface exposed and the type of auxiliary electrode.

The test instrument is GAMRY INTERFACE 1000E. The samples were ground and polished with sandpaper and 1.5 micron diamond polishing paste, and were washed with ethanol and dried. Three-electrode system was adopted, the reference electrode was saturated calomel solution, and the auxiliary electrode was platinum electrode. The working electrode is the tested sample with an effective area of 1cm2. The corrosive medium used is m-SBF  (pH7.4) solution.

m-SBF (pH7.4)  instead of m-SBF

What is the pH of the SBF used?

The pH of the SBF is 7.4

Results

The discussion section is missing. The authors have 2 options: a-rename this section Results and discussion and improve the section by comparing the results with other studies or b- introduce a new section name Discussion.

We renamed this section, Discussion instead Result

Figure 2 is not clear. Please replaced it.

It has been replaced.

Line 214. SEM and EDS images are not diagrams. Please rewrite.

It has been rewrited.

Lines 214-217. The authors are present in the text figure 3 and the next figure 5. The figure must be presented in the order which appears in the text.

The figure has been completely adjusted and revised.

Figure 3. Remove the (a), (b), (c)…. Before the name of the element from the figure. Also, in figure (a) please introduce figure labels in order to highlight the interest zones. Please increase the size of figure (g).

  1. Figure 3and Figure 4 (after modification) instead of Figure 3 (before modification), the (a), (b), (c)… which before the name of the element from the figure have been removed.

Figure 11 and Figure 12 (after modification) instead of Figure 8 (before modification), the (a), (b), (c)… which before the name of the element from the figure have been removed.

I recommend discussing the results of the XRD making reference to other studies.

When discussing the results of the XRD making reference to “Corrosion behavior of Mg–Mn–Ca alloy: Influences of Al, Sn and Zn”, “Mechanical, invitro corrosion and bioactivityper formance of Mg based composite for orthopedic implant applications: Influence of Sn and HA addition”, ect

How were determined the results from Figure 5? Please write in the Material and methods section.

Added 2.5 Theoretical calculation of alloy properties.

“Based on the fact that there are many uncontrollable factors in the research of new materials, the calculation of related properties was carried out using the CALPHAD Technology, which is well developed at present; therefore, the feasibility of the research was confirmed. The research mainly focused on thermodynamic properties and solidification phases. The theoretical formulas include the thermodynamic principles, the Scheil–Gulliver Model, and the calculation of material properties. This part has been discussed in published papers [40].”

[40] Beiyi Ma, Dongying Ju, Qian Liu. Design, Simulation and Performance Research of New Biomaterial Mg30Zn30Sn30Sr5Bi5 [J]. Coatings, 2022, 12, 513. 

Subsection 3.4. First of all, the potentiodynamic polarization results and EIS results must be discussed in different subsections.

The structure of the paper has been adjusted. The content of 3.4 Electrochemical corrosion performance of Mg30Zn30Sn30Sr5Bi5 alloy is divided into two parts, and the section titles are 3.4.1 Potentiodynamic polarization result analysis and “EIS result analysis”.

Potentiodynamic polarization results: Please introduce a table with the values of current density, corrosion rate, corrosion potential and polarization resistance. And discuss all of them. How was calculated the corrosion rate? Regarding Figure 6, why did the authors put the Nyquist diagram next to the polarization curve?

Table 5. Potentiodynamic polarization result.

Material

Ecorr

icorr

Rp

Corrosion Rate

Mg30Zn30Sn30Sr5Bi5 alloy

-634.7

5.669

4.708e3

2.591

Pure Mg

-845.6

27.28

3.093e2

12.7

Units: Ecorr: mV, icorr:μA/cm-2 , Rp: Ω·cm2, Corrosion Rate: mpy.

The position of the Nyquist diagram has been changed.

Also, why the Nyquist diagram for Figure 7 it is different? And for each sample are the results from Figure 7? Please clearly specify.

Figure 10 instead of Figure 7. Nyquist in figure 10 is the fitting equivalent circuit by Equivalent Circuits software, which is the fitting effect diagram given by the software. In this study, the electrochemical corrosion resistance of three specimens was tested, and the results were biased. In this paper, the best group of results was selected.

The results of the corrosion test must be discussed by comparing the obtained results with others from the literature.

Added the reference [53,54]. The corrosion test was discussed by comparing the obtained results with others from reference [53]-[54].

[53].Zhao Jingmao, Li Tong, Zhao Xuhui. Corrosion Behavior of Four Cast Magnesium Alloys in Simulated Body Fluid [J]. CORROSION SCIENCE AND PROTECTION TECHNOLOGY, 2015, 27(5): 444-448.

[54] Stefano Farè, Qiang Ge, Maurizio Vedani, Gianmarco Vimercat, Dario Gastaldi, Francesco Migliavacca, Lorenza Petrini, Stefano Trasatti. Evaluation of material properties and design requirments for biodegradable magnesium stents [J]. Revista Matéria, 15(2): 96-103.

Subsection 3.5. Please improve the discussion and also add labels to highlight the interested zone in Figures 8(a) and 9.

The highlight zone have been added. Reference: Radha Rajendran, D. Sreekanth. Mechanical, in vitro corrosion and bioactivity performance of Mg based composite for orthopedic implant applications: Influence of Sn and HA addition.

Results and Limitation

This section must be replaced with the Conclusion section.

Conclusion instead of Results and Limitation. Some contents have been modified and highlighted

Round 2

Reviewer 3 Report

The paper can be published in the present form. 

Author Response

Dear Reviewer,

We appreciate you very much for your comments and suggestions.